# DNA Methylation Changes and Phenotypic Adaptations Induced Repeated Extreme Altitude Exposure at 8848 Meters

**DOI:** 10.3390/ijms252312652

**Published:** 2024-11-25

**Authors:** Shixuan Zhang, La Yang, Zhuoma Duoji, Danzeng Qiangba, Xiaoxi Hu, Zeyu Jiang, Dandan Hou, Zixin Hu, Zhuoma Basang

**Affiliations:** 1High Altitude Health Science Research Centre of Tibet University, Tibet University, 10 East Zangda Road, Lhasa 850000, China; sxzhang21@m.fudan.edu.cn (S.Z.); yangla721@utibet.edu.cn (L.Y.); dorjee1979@163.com (Z.D.); ehsanjayden@gmail.com (D.Q.); 2State Key Laboratory of Genetic Engineering, School of Life Sciences & Human Phenome Institute, Fudan University, Shanghai 200438, China; xiaoxihu21@m.fudan.edu.cn (X.H.); jiangzeyu367@gmail.com (Z.J.); houdandan@fudan.edu.cn (D.H.); 3Artificial Intelligence Innovation and Incubation Institute of Fudan University, Shanghai 200438, China

**Keywords:** adaptive evolution, epigenetic modifications, extreme exposure

## Abstract

Repeated extreme environmental training (RET) enhances adaptability and induces lasting methylation modifications. We recruited 64 participants from a high-altitude region (4700 m), including 32 volunteers with repeated high-altitude exposure, reaching up to 8848 m and as many as 11 exposures. By analyzing 741,489 CpG loci and 39 phenotypes, we identified significant changes in 13 CpG loci (R^2^ > 0.8, ACC > 0.75) and 15 phenotypes correlated with increasing RET exposures. The phenotypic Bayesian causal network and phenotypic-CpG interaction networks showed greater robustness (node correlation) with more RET exposures, particularly in systolic blood pressure (SP), platelet count (PLT), and neutrophil count (NEUT). Six CpG sites were validated as significantly associated with hypoxia exposure using the GEO public da-taset (AltitudeOmics). Furthermore, dividing the participants into two groups based on the number of RET exposures (*n* = 9 and 4) revealed six CpG sites significantly corre-lated with PLT and red cell distribution width-standard deviation (RDW.SD). Our findings suggest that increased RET exposures strengthen the interactions between phenotypes and CpG sites, indicating that critical extreme adaptive states may alter methylation patterns, co-evolving with phenotypes such as PLT, RDW.SD, and NEUT.

## 1. Introduction

High-altitude environments are considered extreme, and repeated training in such conditions may enhance exercise capacity. Research has demonstrated that repeated environmental exposure and training can lead to the development of new adaptation patterns, such as improved motor performance [1,2].

Environmental stress and homeostasis significantly influence epigenetic modifications [3,4]. Studies have demonstrated that DNA methylation states can evolve in response to environmental factors and subsequently impact human physiological phenotypes [5,6]. This suggests that DNA methylation plays a crucial role in modulating human functions. The adaptive regulation of homeostasis can effectively suppress stress responses triggered by external environmental factors [7], thereby reducing the physiological burden on the body [7], including enhancing immune system adaptation [8]. Research on methylation in the *LINE-1* and *EPAS1* genes among Andean, Tibetan, and Ethiopian populations further suggests possible environmental adaptations to high-altitude conditions [9]. The interaction between the environment and epigenetic modifications creates a “memory state” [10] that helps counterbalance physiological phenotypes, thereby maintaining the dynamic equilibrium of the internal environment—an essential characteristic of dynamic environmental adaptation. Epigenetic memory serves as a valuable marker for assessing environmental adaptation traits, such as aging status [11]. Additionally, epigenetic memory can function as a chronometric mechanism during extreme training, guiding individuals toward optimal exercise adaptation states [12].

We recruited 64 Tibetan volunteers, including 32 experienced Everest climbers (Figure 1A). In our prior study, we explored phenotypic and methylation differences between climbers and non-climbers, highlighting variables like neutrophil counts, platelet levels, and genes such as *PTEN* [13]. However, we did not examine how climbing frequency affects DNA methylation. In the current study, we focused on whether DNA methylation reflects a “memory” of repeated extreme environmental exposure. We observed significant changes in 15 phenotypes following repeated extreme environmental training (RET). DNA methylation analysis identified 13 CpG sites strongly associated with RET (R^2^ > 0.8, ACC > 0.75), forming a network (Coef. > 0.3) with five phenotypes, including SpO_2_ and PLT. The network’s robustness increased post-RET, indicating adaptive changes. Validation with the AltitudeOmics dataset confirmed these findings, suggesting RET-induced methylation modifications and unique adaptive traits involving genes like *PLCH* and *LIPN*, as well as phenotypes such as PLT and SpO_2_.

## 2. Results

### 2.1. Characteristics of the Variation of CpG under RET Perturbation

Genome-scale DNA methylation was assessed in blood samples from 64 Tibetans, analyzing approximately 741,489 CpG sites. We utilized Methylation Gene Base Analysis (MGBA) and ordinary least squares (OLS) regression to examine the impact of climbing age (CA, the number of summits at 8848 m) on CpG methylation, adjusting for age, BMI, and immune infiltration. Our analysis revealed that 2396 CpGs in 151 genes were significantly associated with CA (Bonferroni ≤ 0.05, Appendix A). To further elucidate the biological pathways affected by these differentially methylated sites, we conducted a Gene Ontology (GO) enrichment analysis, identifying 36 enriched pathways based on the 151 genes.

Among the 151 genes, 13 CpG loci exhibited a strong linear correlation with climbing age (CA), with a coefficient of up to 0.8 (Appendix A) and an accuracy greater than 0.75 for distinguishing between RET and non-RET groups. These CpGs showed significant changes in methylation associated with CA (R^2^ ≥ 0.8, PBonferroni ≤ 0.05, Figure 1B), with 9 of these CpGs located in the open sea region (Appendix A). Notably, none of these 13 CpGs were found in genes upstream or downstream of HIF-1α. Among these genes, *LIPN*, which encodes a lipase involved in lipid metabolism at the skin barrier [14,15], and *PLCH1*, encoding phospholipase C-Eta1, which affects nervous system excitability [16], were highlighted. Additionally, other significant CpGs included cg19931348 (*IP3*), cg08840010 (*TNFRSF9*), and cg21702188 (*STX5*) (Appendix A).

### 2.2. Feature of the Variation of Phenotype under RET Perturbation

To investigate phenotypic changes under RET perturbation, we assessed 39 phenotypes, including 19 hematological, 16 serum biochemical, and 4 physical traits (Appendix A). Of these, 15 phenotypes were significantly associated with RET (*p* ≤ 0.05, |Coef.| > 0.2, Figure 1C). Our previous research indicated that all 32 RETs were within the healthy range for physiological indicators [13]. We observed a significant positive correlation between RET perturbation and SpO_2_ (Coef. = 0.61) and a significant negative correlation with SP (Coef. = −0.66). Additionally, eight hematological phenotypes related to inflammation and hematopoiesis—MCH, RDW.CV, RDW.SD, NEUT.R, NEUT, MCV, PLT, and MCHC—showed significant associations with RET perturbations. In blood biochemical phenotypes, RET perturbation was negatively correlated with TBIL and Cre and positively correlated with TG. Overall, RET perturbation influenced various physiological phenotypes, particularly SpO_2_ and SP, indicating improved adaptive characteristics of the population following repeated RET [17].

To assess the impact of RET perturbation on multi-organ system interactions, we categorized 15 phenotypes into 7 systems based on the literature (Appendix A). A Bayesian causal network analysis (Coef. > 0.3) was conducted on 8 RET populations, resulting in 3 distinct networks (Figure 2A). Small-world network evaluation revealed a significant increase in mutual compensatory effects between these networks as RET perturbations increased (Figure 2B, Robust CA = 8 = 9.88). In summary, repeated RET perturbations appear to enhance compensatory interactions between organ systems, suggesting the potential formation of environmental memory within these systems.

### 2.3. Interaction of CpG and Phenotype under RET Perturbation

To explore the impact of RET perturbation on the interaction between phenotypes and CpGs, we analyzed a Spearman network involving 13 CpGs, climbing age (CA), and 15 phenotypes (Appendix A). Some CpGs showed low associations with certain phenotypes, such as MCH and MCV (*p* < 0.05, |Coef.| > 0.3). We retained the strong network involving 6 phenotypes with high associations to the 13 CpGs (MP-SSN), including PLT, SP, NEUT, NETU%, and SpO_2_ (N Related in CpG ≥ 13, Appendix A). PCoA analysis differentiated between RET and non-RET populations with 13 CpGs and between more than 6 RET exposures with 12 CpGs (Appendix A). To identify the biological gene ontologies influenced by the MP-SSN, we conducted Reactome Pathway Knowledgebase and Gene Ontology (GO) analyses. Among the 13 CpGs (Appendix A), 10 enriched pathways were found. For instance, *LIPN* and *PI3* are involved in keratinization, *LIPN* and *PLCH1* in lipase activity, and *EMR1* and *TNFRSF9* in the outer plasma membrane.

To examine the effects of repeated RET perturbations on interactions within the MP-SSN network, we constructed eight Bayesian causal networks (|Coef.| > 0.3) for the RET population, resulting in three distinct networks (Figure 2C(a–c)). Small-world network evaluation revealed a significant increase in mutual compensatory effects among these networks as the number of RET perturbations rose (Figure 2C(d), Robust CA = 8 = 11.87). Additionally, the impact of phenotypes on CpGs was notably enhanced with increasing RET perturbations, whereas CpGs influenced phenotypes only at 3 RET exposures (Figure 2C(e)).

### 2.4. MP-SSN Network CpGs Cross-Omics Cross-Queue Validation

To validate the role of methylation modifications in key MP-SSN nodes on downstream genes, we used the AltitudeOmics hypoxic cohort to analyze 13 CpGs under hypoxic conditions with Trait Methylation Analysis (eQTM). We identified significant correlations for four genes at different hypoxia exposure points: cg25907743 (*LIPN*) (Figure 3A), cg18623216 (*PLCH1*) (Figure 3B), cg07480762 (*PLCH1*) (Figure 3C), cg25234117 (*PLCH1*) (Figure 3D), cg05025071 (*EMR1*) (Figure 3E), and cg21702188 (*STX5*) (Figure 3F). All these CpGs showed significant hypermethylation in the Return group (Appendix A). Notably, *LIPN* exhibited a significant negative correlation with climbing stages (Figure 3A) and a significant hypermethylation with low expression in the Return group (Coef. = −0.71, *p* = 0.005; Appendix A and Figure 3G(a), *p* ≤ 0.05). For *PLCH1*, three CpG sites showed significant negative correlations at Day 7 (*p* ≤ 0.05) and hypermethylation with reduced expression at both Day 7 and Return (Appendix A and Figure 3G(b), *p* ≤ 0.05). *EMR1* also showed significant eQTM at Day 1 (Coef. = −0.61, *p* = 0.087) with increased methylation in the Return group (Appendix A and Figure 3G(e), *p* ≤ 0.05). Overall, the four CpGs demonstrated a significant positive correlation with increasing RET (Coef. _LIPN_ = 0.55, Coef._PLCH1_ = 0.54).

### 2.5. Features of the MP-SSN Network in Populations with Differences in Oxygen Saturation

To evaluate adaptation to hypoxic environments, we used SpO_2_ as a marker and performed a linear regression analysis between Climbing Age (CA) and SpO_2_ to classify the population into high and low RET adaptation groups (Figure 4A, nhigh adaptability = 28; nlow adaptability = 36). Among the 39 physiological phenotypes, nine showed significant differences between the high- and low-adaptation groups, with higher SpO_2_ and LYM% in the high-adaptation group (Appendix A). Additionally, we observed significant variance changes for five CpGs between the two groups (Appendix A).

Among the SpO_2_-based groupings with at least three volunteers each where CA equals 5, we analyzed the correlations between phenotypes and CpGs within high- and low-adaptation groups. Five CpG sites in the high-adaptation group showed significant associations with multiple phenotypes (Figure 4B). Specifically, these CpGs exhibited a strong positive correlation with RDW.SD (*p* < 0.05). Additionally, SpO_2_ and MXD negatively correlated with cg25907743 (*LIPN*) (*p* = 0.0046, coef. = −0.84), while cg25234117 (*PLCH1*) correlated significantly with various PLT-related indices. Notably, after five RET exposures, methylation levels of these CpG sites differed between the high- and low-adaptation groups (Appendix A), with the high-adaptation group showing higher SpO_2_ and lower levels of RDW.SD, SP, and PLT.

## 3. Discussion

In this study, we recruited 64 Tibetan volunteers living at 4700 m and identified significant correlations between 15 physiological phenotypes and repeated high-altitude training (RET). We observed that RET increased mutual associations among seven physiological systems (Robust_CA_ = 9.89). OLS regression revealed 2396 CpG loci (151 genes) significantly related to RET, with 13 CpGs strongly associated with climbing age (CA) (R^2^ ≥ 0.8, *p*
_Bonferroni_ ≤ 0.05). Bayesian causal networks indicated increased internal interactions with more RET perturbations (Robust_CA_ = 9.90). GO and KEGG analyses showed these genes are involved in processes like lipase activity and keratinization. Validation with the AltitudeOmics dataset confirmed significant eQTM for six CpGs under hypoxia (*p* ≤ 0.05) and highlighted significant methylation changes (*p* ≤ 0.05). Notably, we found significant differences in five CpGs among populations with varying blood oxygen saturation and observed strong correlations between phenotypes such as PLT and RDW.SD in the high-blood oxygen saturation group with CA equal to 5.

High SpO_2_ values and low SP values in the RET population reflect the population’s strong ability to adapt to hypoxia training. The seven systems in which the 15 RET-associated phenotypes were located reflected increased robustness after RET (Robust_CA_ = 8 = 9.90), a result of environmental resistance [5,11]. The ability to rapidly and completely recover from deviations from normal physiological states or damage to the human internal environment caused by environmental stress may be an adaptation at the population level. The increased robustness of organ systems reflects the enhanced environmental resistance of the population after RET, which may be associated with methylation memory. Methylation modifications can be influenced by the internal environment, affecting blood, while epistatic modifications can also affect the phenotype. RET not only advanced the formation of methylation memory but also formed a strong organ system network. We also found a significant correlation between CpG methylation and phenotype. We found that 13 CpG sites in the MP-SSN network maintained higher methylation levels after multiple RET exposures, and the robustness of the network was also significantly enhanced after RET, while the correlation between CpG methylation and phenotype was increased. We suggest that MP-SSN is the key to responding to RET.

High SpO_2_ values and low SP values in the RET population indicate a strong adaptation to hypoxia training. The seven systems encompassing 15 RET-related phenotypes showed increased robustness after RET (Robust_CA_ = 9.90), highlighting improved environmental resistance [5,11]. This enhanced robustness reflects the population’s ability to recover from environmental stress and maintain physiological stability [5,6,7,8,9]. This adaptation may be linked to methylation memory, which influences processes such as blood pressure [12], hematopoiesis [18], and the immune system [19]. RET not only promotes methylation memory formation but also strengthens organ system networks. We observed that 13 CpG sites in the MP-SSN network had higher methylation levels after multiple RET exposures, with increased network robustness and stronger correlations between CpG methylation and phenotypes. Thus, MP-SSN appears crucial for adapting to RET.

MP-SSN networks reveal key connections in RET adaptation. Validation using the database confirmed six CpG sites, such as cg25907743 (*LIPN*), cg25234117 (*PLCH1*), cg18623216 (*PLCH1*), and cg07480762 (*PLCH1*), which are implicated in lipid metabolism and epidermal keratinocyte differentiation. Notably, cg25907743 (*LIPN*) is located in the transcriptional enhancer region, 200–1500 bp upstream of the transcription start site [20,21]. The *LIPN* gene, involved in triglyceride hydrolysis and keratinocyte differentiation, is highly expressed in the skin and lung [22,23]. *LIPN’s* role in maintaining the epidermal barrier is crucial, as impaired lipid metabolism can disrupt skin permeability and protection [24,25,26,27]. Hypoxic conditions can hinder fatty acid synthesis and barrier recovery [28]. Additionally, *LIPN* is vital for skin barrier restoration and has been linked to pulmonary health [20]. Our results showed increased methylation of cg25907743 (*LIPN*) in the RET group, which likely downregulates LIPN expression, affecting barrier repair and aligning with hypoxia-related skin and lung damage [29,30]. Furthermore, while cg19931348 (*PI3*) showed no significant eQTM changes under hypoxia, its gene Elafin, which inhibits neutrophil elastase, may contribute to RET adaptation by protecting against lung injury [31,32,33,34].

The MP-SSN network reflects a significant increase in interactions between phenotypes and CpGs with the number of RET exposures, indicating that repeated exposure leads to stronger methylation-phenotype interactions, which characterize resistance to environmental changes. Within the MP-SSN network, genes such as *ACSL3*, *PPBPL1*, and PI3 are affected by SpO_2_ during multiple exposures. *PPBPL1* is a platelet-related pseudogene without biological significance. At five RET exposures, SpO_2_ is at its highest, and the hypermethylation of *ACSL3*’s cg23325384 may inhibit its expression, leading to increased fat storage, consistent with lipid accumulation under hypoxic conditions [35,36]. At eight RET exposures, SpO_2_ decreases, and the hypermethylation of *PI3*’s cg19931348 may inhibit its expression; *PI3* is significantly downregulated in patients with acute respiratory distress syndrome [32]. Overall, five RET exposures may represent an optimal adaptive state beyond which damage may occur.

More organ systems are significantly affected in extreme environments, and we found that the nervous system may also be subject to strong environmental selection. Studies suggest that phospholipase C-eta (PLC-η) signaling may play a fundamental role in the brain [37,38,39]. *PLCH1* is a PLC-η family member whose three CpGs are located in the transcriptional enhancer and promoter regions (opensea TSS1500, 5’UTR, and TSS200). The FANTOM5 database suggested that the *PLCH1* gene is expressed at high levels in brain and lung tissues, and its expression is higher in nerve fibers, respiratory ciliated cells, and alveolar type 2 cells. In addition, the UniProtKB database reveals the protein is distributed in the plasma membrane. PLC-η is abundant in the brain [37,38,39], it is the most Ca^2+^-sensitive enzyme, and it may participate in the G protein-coupled receptor (GPCR) signaling pathway [37,40], which regulates neuroexcitability in the brain and the endocrine system through regulating intracellular Ca^2+^ dynamics and protein kinase C activation [37]. Hypoxia increases the sympathetic excitatory effect in plain populations [41], unlike high-altitude Indigenous populations, where sympathetic excitability is significantly reduced [42]. Altered neural excitability may regulate organ states. Compared with non-RET-exposed controls, the methylation level of cg25234117 (*PLCH1*) was significantly increased in RET-exposed individuals, and the validation results suggest that it may downregulate the expression of *PLCH1* to weaken neural excitation. Consistent with previous studies, multiple exposures at high altitudes consistently weaken neuronal excitation in high-altitude populations [42].

The discussion highlights that CpGs related to *LIPN* and *PLCH1*, which are involved in the epidermal barrier and nervous system, might play a role in adapting to RET. This adaptation could be linked to the enhanced light environment during RET exposure [43] (Figure 5). At high altitudes, lower concentrations of atmospheric molecules such as O_2_ and CO_2_ result in reduced UV absorption, which can damage the skin barrier [44]. Additionally, high-altitude areas experience intense illumination and prolonged daylight in the visible spectrum (380–780 nm). Extreme light conditions significantly impact photobiological processes mediated by ipRGCs through the human eye [45,46], influencing circadian rhythms, mood, cognitive behavior, and other central nervous system functions [47].

Repeated exposure to Mount Everest’s extreme light conditions may trigger responses in the body. Bright light can activate the sympathetic nervous system, increasing platelet reactivity and affecting stress hormone release, which in turn influences the myocardial response to ischemia and hypoxia [48]. Moreover, high light exposure upregulates microRNAs, enhances glycolysis, and promotes cardioprotection via circadian rhythm proteins (e.g., Period 2) [49]. Consistent with these findings, the downregulation of cg25234117 (*PLCH1*) correlates with changes in PLT and HR in the RET population, suggesting that extreme light conditions impact neurological regulation.

On the other hand, hypermethylation of cg25907743 (*LIPN*) may facilitate skin barrier repair, aiding adaptation to harsh light environments. In summary, our results suggest that prolonged RET exposure leads to light-adaptation changes, such as enhanced skin barrier repair and increased nervous system excitability. However, further experimental validation is needed to confirm these biological mechanisms.

We highlight the potential applications of understanding methylation modifications and phenotypic interactions from repeated extreme environmental training (RET). These findings can inform health monitoring protocols for high-altitude climbers, athletes, and military personnel, enhancing adaptability and health outcomes. Identifying predictive biomarkers for hypoxia adaptation allows for better assessment and personalized training programs. The significant correlations between specific CpG loci and phenotypes like PLT, RDW.SD, and NEUT provide insights into gene-environment interactions, benefiting athletic performance and public health strategies. Our research lays the groundwork for future studies on epigenetic adaptations and potential therapeutic interventions.

The study has several limitations. Due to constraints in sample collection, we were unable to conduct long-term follow-up on the climbers, which restricted our ability to explore the precise temporal changes in methylation modifications. Time as a variable remains a potential limitation of our research. Additionally, while our study identifies correlations between methylation modifications and climbing exposure, it primarily offers intriguing data-driven observations without experimental validation. Future research will address this gap by employing cellular and animal models to validate these findings and further elucidate the causal relationships. Furthermore, the intermittent use of supplemental oxygen by climbers during expeditions, though not continuous, could not be accurately quantified to adjust our data, potentially influencing methylation states. However, none of the participants engaged in long-term oxygen therapy outside of climbing activities, mitigating its impact on our overall findings. Despite our efforts to elucidate the potential links between phenotypes and CpGs in the MP-SSN network and the adaptive traits from repeated extreme exposure, the lack of extensive data and experimental validation prevents us from asserting the absolute accuracy of our findings.

## 4. Methods

### 4.1. Study Group and Phenotype

Our study, involving blood samples from 64 Tibetan male participants, was approved by the Tibet University Ethics Review Committee [13], with informed consent obtained. This included 32 individuals who underwent repeated extreme environment training (RET), with an average age of 33.25 years, and 28 of whom had ascended Mount Everest (8848 m) more than five times. All subjects were born in Shigatse, Tibet, at an altitude above 4700 m. The control group consisted of 32 non-RET males, with an average age of 33.72 years, engaged in physical labor in Nagqu, also above 4700 m.

We selected the control group (non-RET) based on the genetic background of the climbers (Tibetan, China, 4700 m altitude, with three generations living at this altitude). All RET participants were professional high-altitude guides or climbers with careers spanning over 15 years, ensuring a consistent genetic background as they had all lived and trained above 3760 m. To account for the potential confounding effects of long-term high-altitude adaptation, we matched the RET group with a control group of individuals living in similar high-altitude environments without professional high-altitude physical labor. Furthermore, we considered the possibility of inherited methylation modifications and ensured that the ancestry of the participants had no prior involvement in mountaineering or high-altitude rescue activities.

To avoid interference, we excluded factors such as education level, gender, age, smoking, and drinking. Blood samples were collected one month after the Everest expedition and processed under identical conditions. We analyzed 39 phenotypes, including 19 hematological phenotypes (Appendix A). This careful selection of participants ensured that our comparisons would be as precise and meaningful as possible, isolating the specific impacts of extreme environment training from other potential influences.

### 4.2. Methylation Array Processing and Cell Proportion Correction

In this study, using the Illumina Methylation EPIC array (850K), we identified methylated regions of 64 samples. CHAMP [50] Package provided by R (Version 3.6.3) language was used for quality control (QC), standardization, and batch correction. We set the absolute value of b > 0.1 to screen the differentially methylated probes (DMPs). Furthermore, we set the absolute value of b > 0.05.

Variations in cell types, health behaviors, and population structure subgroups can impact the accuracy of DNA methylation data. To address these potential confounders, we applied linear regression for adjustment during the methylation difference analysis. Initially, we used a deconvolution [51] approach to assess the proportions of various cell types between RETs and non-RETs (Appendix A), identifying differences in neutrophil cells, natural killer cells, and CD4 T cells between the two groups (Appendix A). Finally, to mitigate the impact of individual DNA methylation differences caused by variations in blood cell proportions, the study employed cell infiltration scores as a correction factor. These scores were used as covariates to adjust subsequent analyses. These analyses were conducted using R (v 4.0), with cell infiltration scores calculated via the EpiDISH package [52].

### 4.3. Methylation Gene Base Analyses (MGBA)

The study set the Beta value of the methylation modification of the ‘*t*’ gene in the ‘*i*’ individual based on the smoothed functional principal component analysis (FPCA) [53] method to: $$x_{i}\left(t\right)=\left[0,1\right]\;\;i=1,\ldots ,\;n$$
*n* is the number of samples. Each Beta value data is projected into the finite-dimensional space of the FPC or eigenfunction by FPCA. Afterward, let $\beta_{j}(t)$, *j* = 1,2,… be a set of FPCs(obtained by solving the integral characteristic equation, Equation (1)):(1)$$\varepsilon_{ij}=\int_{T}x_{i}\left(t\right)\beta_{j}\left(t\right)dt,$$
The formula where $\varepsilon_{ij}$ is the FPC score of the ‘*i*’ individual (for its calculation, refer to the book: Functional data analysis [54]).

Following this, similar to the Fourier series or wavelet expansion, we construct the genotype spectrum function $x_{i}\left(t\right)$, which can be expanded according to the orthogonal FPC [53], where the FPC is used as the basis function.
(2)$${\;x}_{i}\left(t\right)=\sum_{j=1}^{J}\varepsilon_{ij}\beta_{j}\left(t\right),\;$$

In the study FPC was constructed from the Beta contained in a single gene, and each sample had the FPC score of all genes. As a result, the FPC score effectively downscales and compresses the gene methylation modification information [55], which helps to observe the overall gene modification degree in a macroscopic way. Based on the FPC scores, the study looked for genes with large methylation modification differences between RETs and non-RETs by two-sided test and used the Bonferroni [56] method for *p*-value correction, setting a threshold of P Bonferroni < 0.05.

### 4.4. Association between Methylation and Phenotype and Number of Climbs

#### 4.4.1. Screening of Climbing Age-Related CpG

The study used climbing age as the independent variable and CpG sites in differential genes in MGBA as the dependent variable, applying ordinary least squares (OLS) regression [57]. OLS was adjusted to account for potential confounders such as actual age, BMI, and population structure. CpG sites most strongly influenced by climbing age were selected using an R^2^ ≥ 0.8 as the screening criterion. Additionally, covariate adjustments for immune infiltration degree and population structure were included. Based on CpG loci significantly correlated with climbing age (OLS), a classification model was constructed using nonnegative matrix factorization (NMF) and K-Means (KM) clustering, with the optimal classification identified by the fastest cophenetic decline. The classification effectiveness of CpG sites was evaluated using ACC, calculated as follows:$$ACC=\frac{right}{all}$$
In the formula, “ACC” denotes the number of correct classifications, and “all” represents the total number of classified samples. These calculations were performed using R (version 4.0), utilizing the ‘NMF’ package and the ‘kmeans’ function.

#### 4.4.2. Screening of CLIMBING AGE-Related Phenotype

This section investigated the correlation between climbing age and physiological phenotypes using Spearman’s algorithm. We filtered for phenotypes significantly associated with climbing age (*p* ≤ 0.05, |Coef.| > 0.2). These analyses were conducted using R (version 4.0).

### 4.5. Network between Methylation and Phenotype

Explore MP-SSN: The study employed the Spearman algorithm to construct the Strong Synergistic Network of Methylation and Phenotype (MP-SSN), focusing on differential physiological phenotypes, climbing age, and significant CpG sites identified through OLS (|Coef.| > 0.3, *p* ≤ 0.05). Additionally, we assessed the classification effectiveness of various CpG and phenotype combinations across different climbing age populations using Principal Coordinates Analysis (PCoA). These analyses were conducted in R (version 4.0) with the ‘vegan’ and ‘ape’ packages, and PCoA plots were generated using the ggplot2 package to illustrate the mean values of the distance matrix and the 95% confidence intervals.

MP-SSN Gene Ontology Study: Conversion of Gene IDs by H. sapiens Entrez gene IDs (1 February 2021) for genes in the MP-SSN network. After that, enrichment studies were performed using the GO (BP\MF\CC) KEGG database (*p* ≤ 0.05).

Detailing MP-SSN: Spearman models may ignore changes in details, so the study used Curve weighted sliding fitting (CWSF) to smooth the CLIMBING AGE series, i.e., calculating a sample-weighted average of all observations within its window at each age. After correcting for the effects of age and BMI on methylation modifications, CWSF was performed using the residual matrix.

### 4.6. Causality Network

Causal network construction: Causal models elucidate key principles such as multiple causation and interactions between component causes [58]. To investigate how the external environment influences the interaction between internal environments, the study employed a causal inference algorithm. The core of our causal networks is based on Bayesian network analysis (BN) [57,59,60,61], which uses graphical models with joint probabilities (directed acyclic graphs) to describe the conditional dependencies and independencies of variables [62]. We constructed BN networks for each of the 15 climbing age-related phenotypes and the MP-SSN. To simplify these BN networks, we applied the Group Lasso algorithm (lambda < 0.3) for network pruning. All calculations were performed in R, utilizing the hc() and bn.fit() functions, and the ‘bnlearn’, ‘flare’, and ‘igraph’ packages.

Causal network evaluation: To objectively assess the condition of organ networks and internal connections within the MP-SSN under varying RET conditions, the study evaluated the causal networks using small-world network features, focusing primarily on robustness [63]. Robustness, in this context, denotes the stability of the network, with higher robustness indicating greater stability. This concept measures the network’s resilience to the removal of connections and assesses the overall impact on network integrity.

### 4.7. Database Verification

To validate the gene change characteristics within the MP-SSN under hypoxic conditions, we utilized the AltitudeOmics dataset from GEO, which includes expression data (GSE103927) and methylation data (GSE105124). This dataset records the responses of 21 participants exposed to high altitude (5260 m) for 16 days, followed by a descent to 1025 m and a return to 5260 m. While the exposure pattern differs slightly from our repeated exposure cohort, it still represents an elevation-increasing exposure pattern. We first assessed the expression Quantitative Trait Methylation (eQTM) properties of genes associated with CpGs in the MP-SSN [64]. Additionally, we performed two-sided tests to identify significant differences in methylation modifications and gene expression across multiple groups (*p* < 0.05).

### 4.8. High and Low Adaptation Differences

Blood oxygen saturation (SpO_2_) [17,65] directly reflects an individual’s oxygen delivery efficiency in high-altitude environments and is associated with the risk of altitude sickness. Tibetan populations who have lived in the Tibetan Plateau for extended periods typically exhibit higher SpO_2_ levels, indicating genetic adaptation. Therefore, we used SpO_2_ characteristics to classify and explore the adaptive differences in DNA methylation modifications in the RET population.

In the study, a linear regression equation between climbing age and SpO_2_ was used to classify 64 individuals into high and low adaptation groups, with high SpO_2_ indicating high adaptation. To account for potential fluctuations in methylation modification effects across different RET stages, we compared methylation modification levels between the two groups using an F-test. Differences in phenotypes between the high and low adaptation groups were analyzed with a *t*-test, with phenotypes standardized for better graphical representation. Finally, to examine significant correlation differences of 13 CpGs with 32 phenotypes under varying RET states, Spearman’s algorithm was used to test the correlation of corrected Beta values (*p* < 0.05, |Coef.| > 0.3).

### 4.9. Statement of Ethical Approval

The methods in this study adhered to relevant guidelines and regulations, including the Helsinki Declaration. All experimental protocols and procedures were approved by the China Tibet University Ethics Committee, ensuring compliance with ethical standards and the protection of participants’ rights. Data collection and use were conducted with the informed consent of all volunteers, who signed consent forms. (Ethic Committee Name: Tibet University Medical Ethics Committee; Approval Code: ZDYXLL2024002; Approval Date: 23 May 2024).

## Figures and Tables

**Figure 1 ijms-25-12652-f001:**
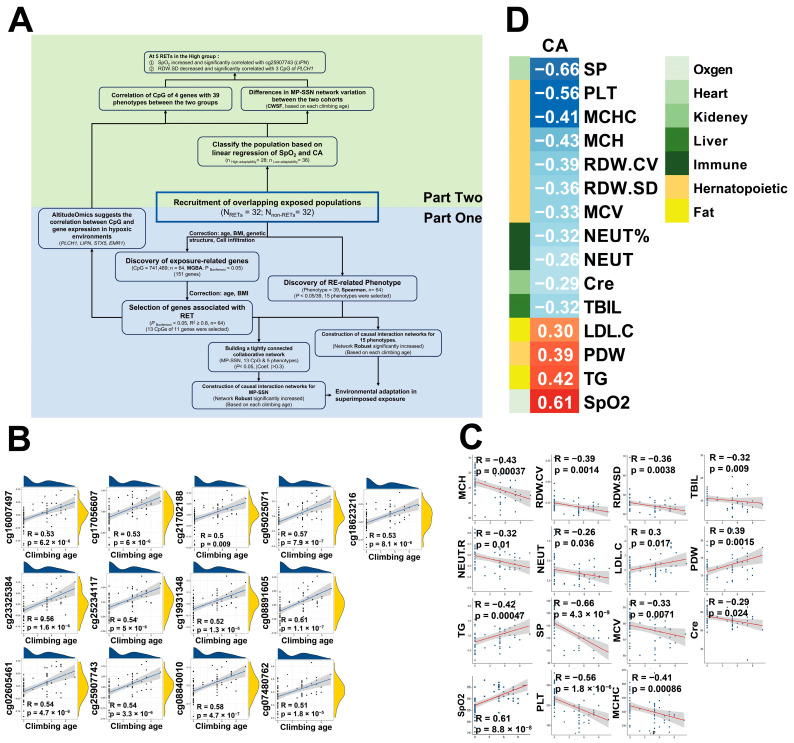
(**A**) The overall research strategy is divided into two parts: First, we explore the relationship between climbing age (CA) and the “memory” of DNA methylation modifications. Second, we categorize all samples into high and low adaptation groups based on SpO_2_ levels and investigate the phenotypic associations of key CpGs within these different adaptive groups. (**B**) The 13 CpGs of 11 genes were significantly and positively correlated with RET perturbations (R^2^_OLS_ ≥ 0.8). (**C**) The 15 phenotypes were significantly correlated with RET perturbations (P Spearman < 0.05). (**D**) Correlation between CA and Phenotypes.

**Figure 2 ijms-25-12652-f002:**
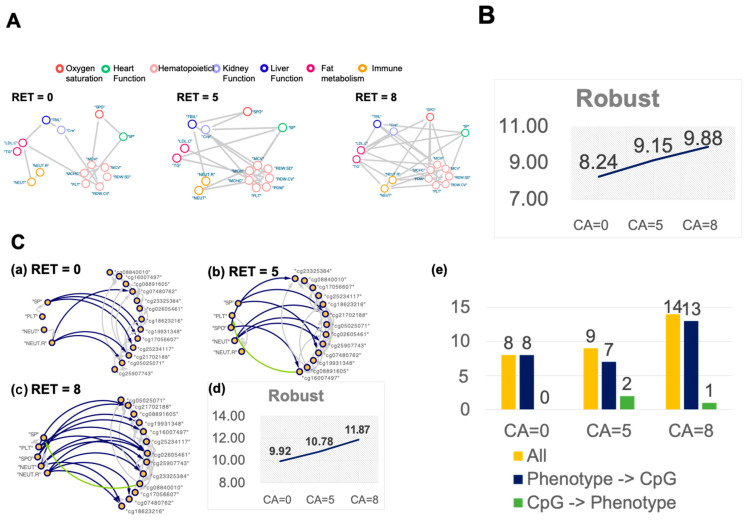
Bayesian causal networks to explore the network properties of organ systems and MP-SSN in different times of RET environments. (**A**) The study constructed Bayesian causal networks for seven organ systems in which 15 physiological phenotypes were located. (**a**–**c**) The networks of RET are constructed at 0, 5, and 8, while the networks of other RETs are not inherently connected. (**B**) Small-world networks were used to evaluate 15 phenotypes of networks with different RET counts, which suggested enhanced network connectivity. (**C**) The network between Phenotype and CpG in different RET frequencies. The green line represents “CpG -> Phenotype”; the blue line represents “Phenotype -> CpG”. (**a**) MP-SSN Bayesian causal network for 0th RET perturbation population. (**b**) MP-SSN Bayesian causal network for 5th RET perturbation population. (**c**) MP-SSN Bayesian causal network for 8th RET perturbation population. (**d**) Small-world networks were used to evaluate MP-SSN networks with different RET counts, which suggested enhanced network connectivity. (**e**) The number of Bayesian causal network connections between CpG and phenotype in the MP-SSN network. “All” in the figure: the total number of connections between the two; “Phenotype -> CpG”: the direction of the Bayesian causal network is that phenotype affects CpG.

**Figure 3 ijms-25-12652-f003:**
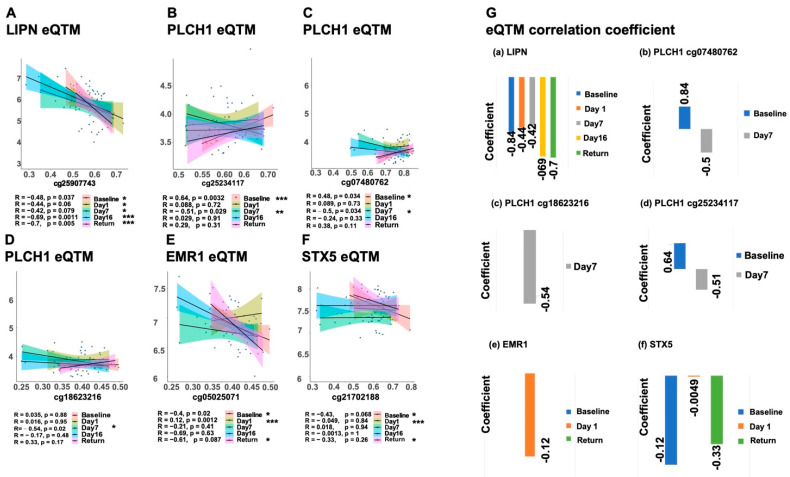
(**A**–**F**) Status of LIPN, PLCH1, EMR1, STX5 genes in the AltitudeOmics. Linear correlation between CpG and gene expression at different stages of hypoxia exposure. (**G**) The values in the graph represent the linear correlation coefficients between each CpG site and gene expression in the A-plot (Coef.). In the figure, * indicates *p* < 0.05, ** indicates *p* < 0.01, *** indicates *p* < 0.001.

**Figure 4 ijms-25-12652-f004:**
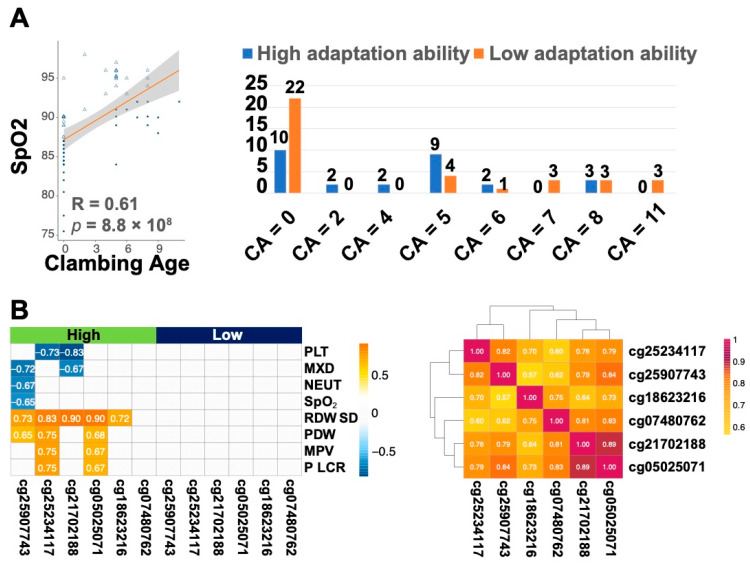
(**A**) A linear relationship between blood oxygen saturation and CA; Number of high and low adaptive populations distributed in different CAs. (**B**) The 13 CpGs are related to 32 Phenotypes in High and Low. All phenotypically unrelated CpGs were excluded from the figure. In addition, the part of the graph without color means no significant correlation (*p* < 0.05, |Coef.| > 0.3).

**Figure 5 ijms-25-12652-f005:**
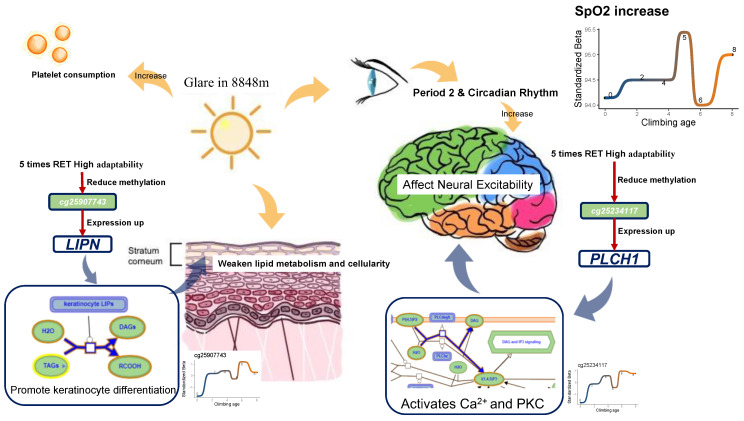
Possible speculation on the effect of *LIPN* and *PLCH1* on blood oxygen saturation.

## Data Availability

Our study will provide a matrix of Beta values of methylation modifications in climbers. In addition, please contact the corresponding author of our paper, Zhuoma Basang, at bsangzm@163.com to obtain permission for data usage.

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
