# Peer review of "DNA Methylation Changes and Phenotypic Adaptations Induced Repeated Extreme Altitude Exposure at 8848 Meters"

_ijms, 2024, doi:10.3390/ijms252312652_

Round 1

Reviewer 1 Report

Comments and Suggestions for Authors

The authors of the study explore the intriguing concept of the impact of repeated exposure to extreme altitudes on changes in DNA methylation. They observed RHT-induced modifications in DNA methylation as well as changes in adaptability to hypoxic environments at the level of LIPN and PLCH1 gene expression and phenotypic traits like platelet count and oxygen saturation.

The study was conducted on a relatively small cohort comprising 32 individuals who underwent ascent to extreme altitude and 32 control individuals. However, considering the highly specific demands on the participats undergoing the intervention, I consider the presented group sizes as sufficient.

The methodology used is adequate, but its presentation, as well as the overall presentation of the results, is quite complicated, which detracts from the clarity and coherence of the article. We recommend reconsidering the chosen method of presenting the results. I have reservations about the study design. If the study monitors possible changes in DNA methylation depending on (repeated) exposure to extreme altitudes, as a control group, instead of the general population, I would directly choose individuals undergoing the intervention before the intervention. Possible differences between the control and intervention branches should also be considered, given the different regions of origin.

To improve the clarity of the article, we definitely recommend adding a section on abbreviations and symbols, as the text uses a very large number of abbreviations, making it considerably confusing.

The study does not mention potential directions in which the results could be used. I strongly recommend adding this section.

The study only marginally mentions that it has several limitations. Therefore, I would ask the authors to add which limitations they consider the most significant.

I suggest the authors adjust the chosen title of the publication so that it is clear that it involves repeated exposure to extreme altitude, as currently, this is only implied by the expression "(8848m)".

Formal shortcomings:

Line 3: Space between words "exposure" and "(8848m)" is missing.

Line 14: The authors indicate that three authors contributed equally to the work, but only two authors are listed.

Lines 88-89: The publication "Functional data analysis" is not cited in the bibliography, and it is unclear what publication is referred to. Please correctly cite this publication.

Line 200: I recommend the authors rephrase/add to the section "...with 9 located in ..." to clarify that they mean 9 CPGs with significant changes in DNA methylation in connection with climbing age (CA).

Line 372: Separate the numeric references "4344".

Line 374: Correct the numeric references "4645".

Figure 3-G: "EMR1" should be indexed as "e" and "STX5" as "f".

Questions for the authors:

1) Do the authors have information on the employment of participants in the RHT branch of the study? Did they consider this knowledge when designing the study or evaluating the results? I believe that in the case of professional high-altitude guides, who frequently move in extreme altitude environments, the observed changes may not be related to a few ascents to 8848m but may rather be due to long-term adaptation of the body.

2) Do the authors have information on the ancestors of the study participants? Could the changes in DNA methylation in participants in the RHT branch also be conditioned by the fact that, for example, their parents may have been adapted to extreme altitude conditions? I believe that in the case of high-altitude guides, it is somewhat possible that one of the parents was also engaged in this profession.

3) In connection with points 1) and 2), I would point out the not entirely appropriately chosen study design. I would expect the chosen model when comparing changes between the general population and a population long-term influenced by exposure to (extreme) altitude. When focusing on changes conditioned by a few ascents to extreme conditions, or changes related to the number of ascents, I would confront these changes with the state before the ascent. Of course, the observed changes may correctly reflect the observed intent of the authors, but a correctly chosen study design would allow for much more straightforward interpretation of the results. I would therefore ask the authors to clarify the chosen study design and also explain why they did not include individuals from the same region as the participants in the intervention branch in the control branch.

4) The authors do not provide information on whether the study participants ascended to extreme altitude with the help of an oxygen mask or without its use. I assume this fact can significantly impact changes in DNA methylation. Could the authors provide this information?

5) Did the authors analyze or consider what might underlie the various adaptability to hypoxic training? The authors state that MP-SSN is key in the response to RHT, but the background of the various adaptability remains unclear.

6) I do not question the interesting observations and conclusions of the authors; however, I was surprised that despite the demonstrated high expression of LIPN and PLCH1 in lung tissue, which is one of the most exposed tissues to extreme conditions during ascent to high altitude, the authors only marginally address this issue. Did the authors also analyze possible changes in the expression of LIPN and PLCH1 in the lungs? Naturally, the question arises whether and, if so, what are the changes in LIPN expression in this location during (repeated) stays at extreme altitude – again under various conditions of ascent (comment 4).

7) Do the authors plan to continue in monitoring methylation levels in the intervention branch? It would undoubtedly be interesting to verify whether the induced changes are long-term or whether, in individuals who no longer undertook further ascents to extreme altitude, the induced changes in DNA methylation are permanent or reversible.

Reviewer 2 Report

Comments and Suggestions for Authors

Review : DNA methylation modifications formed through repeated extreme exposure (8848m) and their phenotypic adaptive evolutionary characteristics

In their manuscript, the authors present results on the effects of repeated extreme environmental training (RET) on clinical phenotypes and try to link them to DNA methylation changes.
The experimental setting is very interesting and the potential results relevant for researchers of different communities. However, besides a number of formal issues (see below) there is a major problem with the approach.

The authors analyze no further specified ‘blood samples’. Thus, independent of whether these are PBMC or whole blood samples, they analyze a composite of different cell types. DNA methylation of CpGs differs between these types. For illustration, I have attached a heatmap for the beta-values of the 13 CpGs highlighted in the study in different blood cell types.
Accordingly, changes in the overall methylation refer either to ‘real’ changes (with potential impact on gene expression) or simply to changes in the blood cell composition.  Blood composition has not been analyzed. Thus, all conclusions regarding changes of DNA methylation and gene regulation are questionable.
Clearly, a subsequent analysis cannot be performed. I suggest applying public available deconvolution methods on the DNA methylation data to quantify cell composition.     

In addition, there are several formal issues. Examples:
The platform and protocol used to measure DNA methylation are not specified.
The term ‘adaptability’ is not explained and not referenced.
Fig. 1A is not cited in the text, and not readable. The explanation given in the caption is insufficient.
Abbreviations (e.g. phenotypes) are not explained.
Etc.

In the current version, I cannot recommend the manuscript for publication.

Round 2

Reviewer 1 Report

Comments and Suggestions for Authors

I would like to thank the authors for the time and effort invested in revising the manuscript, as well as for the thorough responses to the comments, which were intended to fine-tune the work and clarify certain aspects to ensure that the results are presented in a clear and credible manner.

I fully recognize the complexity of conducting experimental research with this focus. Therefore, I express my appreciation to the authors for their mastery of the subject matter, their responsible approach in obtaining and interpreting the results, and I believe that continuing this research holds great potential to yield interesting and valuable outcomes.

Author Response

Dear Reviewer,

We sincerely appreciate your thoughtful feedback and kind words. Your recognition of the effort and care we put into revising the manuscript means a great deal to us. We are particularly grateful for your acknowledgment of the complexity of our experimental research and your encouragement to continue in this line of inquiry. Your insightful comments have been invaluable in refining our work, and we are committed to pursuing this research further with the same level of rigor and dedication.

Thank you once again for your support and guidance.

Sincerely,
Zhuoma Basang

Reviewer 2 Report

Comments and Suggestions for Authors

The authors largely improved the manuscript, e.g. by including details on blood composition and providing information about DNA methylation experiments.

Thereby, it became obvious, that the study is just a further utilization of data of a former study (ref. 13) which includes already all the methylation data and phenotype relations. Despite this close relationship the former study is cited only once in the ‘study group and phenotype’ section of the methods chapter.  This gives the misleading impression that it includes just a detailed description of the cohort.

The new figure 1A now explains the new study very well: ‘First, we explore the relationship between climbing age (CA) and the ‘memory’ of DNA methylation modifications. Second, we categorize all samples into high and low adaptation groups based on SpO2 levels and investigate the phenotypic associations of key CpGs within these different adaptive groups.’

This largely reduces the novelty and the potential interest to readers. In case of publication, it will be essential that the results of the former study are mentioned in the ‘Introduction’.
‘ In a former study, we recruited 64 Tibetan  volunteers, ….’ Etc.

Additional problems:
Although accepted by JGG in 2021 (Basang, Z. et al. Correlation of DNA methylation patterns to the phenotypic features of Tibetan elite alpinists in 546 extreme hypoxia. J Genet Genomics 48, 928-935,) the data of the study seem to be not public. I recommend to retain publication until the methylation data have been uploaded.

The data used for validation purposes (expression data: GSE103927, methylation data: GSE105124, both uploaded in 2017) still miss a citation with unclear reason. I would request a quality assessment.
